# Pulsed Electromagnetic Field Stimulation in Bone Healing and Joint Preservation: A Narrative Review of the Literature

Gaetano Caruso [1] , Leo Massari [2], Sebastiano Lentini [1], Stefania Setti [3] , Edoardo Gambuti [1] and Achille Saracco [1,*]

1 Department of Neurosciences and Rehabilitation, University of Ferrara, c/o "S. Anna", Via Aldo Moro 8, 44124 Ferrara, Italy; crsgtn@unife.it (G.C.); seb.lentini@gmail.com (S.L.); gambutiedoardo@gmail.com (E.G.)
2 Department of Translational Medicine Romagna, University of Ferrara, c/o "S. Anna", Via Aldo Moro 8, 44124 Ferrara, Italy; msl@unife.it
3 IGEA SpA—Clinical Biophysics, Via Parmenide 10/A, 41012 Carpi, Italy; s.setti@igeamedical.com
* Correspondence: achille.saracco@ospfe.it

**Abstract:** *Background:* Biophysical stimulation therapy, Pulsed Electromagnetic Fields (PEMFs) and Capacitively Coupled Electric Fields (CCEFs) have significantly increased in the last twenty years. Due to this, it is necessary to have clear information regarding their efficacy, therapeutic indications and expected objectives. *Application fields:* There is a unanimous opinion regarding the usefulness of applying biophysical therapy on the bone compartment both in terms of the tissue-healing process and the symptoms associated with this situation. Differently, but no less important, positive results were observed in the joint compartment, especially with regard to the inhibition of the inflammatory process. Good results for chondroprotection were obtained in vitro and after a surgical procedure. New studies have shown the effectiveness also in cases of osteoporosis. *Conclusions:* The effectiveness of PEMFs and CCEFs on the bone-healing process and on joint preservation in the orthopedic and traumatology fields has consolidated evidence in the literature. We have also found positive results for symptoms and patient compliance with rehabilitation therapies. Therefore, their notable applications can be envisaged in the fields of prosthetic surgery and sports medicine.

**Keywords:** pulsed electromagnetic fields; PEMFs; capacitively coupled electric fields; CCEFs; bone healing; joint preservation; traumatology; sport medicine; hip and knee replacement; reverse shoulder arthroplasty





## 1. Introduction

Pulsed electromagnetic fields (PEMFs) bring about a biological response by directly inducing electric currents in the target area. This treatment was introduced by Bassett et al. in 1977 [1]. In 1979, PEMFs were approved by the Food and Drug Administration. Subsequently, this therapeutic approach has been used with increasing success in the orthopedic field. In fact, the musculoskeletal system is highly sensitive to such stimuli. Clinical biophysics exploits the effect of PEMFs on biological systems to obtain benefits during different biological processes.

Over the last twenty years, the approach to this discipline has changed considerably, leading to a clear improvement in the physical parameters relevant to these processes with consequent optimization of therapies and clinical effects.

Biophysical stimulation can reduce potential risks by promoting osteogenic stimulus and reducing healing times. There are numerous publications in the literature in favor of the use of these physical therapies in risk conditions; however, they must be used following the correct administration criteria [2].

The possibility of delivering the therapy locally allows you to optimize the treatment while avoiding any dose-dependent side effects. This characteristic, which makes it potentially suitable for chronic therapies, excludes its use for systemic pathologies [1].

The use of biophysical therapy in bone tissue pathologies is now widespread, and, consequently, there is possibility for evaluating large scientific production in this regard. More recently, biophysical therapy has been extended to joint tissue and related pathologies.

The right combination of physical parameters and the right daily dosage are fundamental for the quality of the therapy provided and impact the peak amplitude of the signal, frequency, and shape of the impulse generated using the devices. Optimizing these parameters translates into an improvement in the dose–response of the therapy and, therefore, in the effectiveness of the treatment. For this reason, the use of equipment supported by verified publications and data is essential.

A better understanding of the mechanisms of interaction of biophysical stimuli on the cellular component has given encouragement for their use in the clinical field, with the application of PEMFs/CCEFs designed to reap benefits in the process of repair and regeneration of bone and cartilage tissue.

PEMFs act on tissue in two ways. First, the magnetic field creates a force on the molecules present in the tissue, which depends on their magnetic reactive properties. In addition, the induced electric field exerts a force on the ions present in the tissue [3]. The combination of these two mechanisms results in a forced movement of ions or charged particles, such as proteins [4]. Low-frequency fields appear to have a greater bioactive potential than static magnetic fields. Moreover, a pulsed magnetic field is twice as effective as a continuous one [5]. These changes are mediated by an increase in the endogenous production of growth factors for the bone and cartilage tissue with a consequent increase in the activity of the stem component obtaining greater and more rapid tissue regeneration.

CCEF devices generate an electric field by storing charge on two parallel electrode plates. One electrode is positively charged and the other stores a negative charge, forming an electric field between them. The intensity of the field is inversely proportional to the distance between the electrodes. Several studies have shown CCEFs promote cell differentiation, proliferation, morphology, adhesion, migration, and function [6].

### 1.1. Modulation of Membrane Receptors for Adenosine

Adenosine and its receptors play a fundamental role in cellular homeostasis; through the expression of the $A_{2A}$ receptor, it regulates the pathway Wnt/$\beta$-catenina which is fundamental in anabolic processes at the bone and cartilage level [7]. Exposure to electromagnetic fields causes greater exposure of the $A_{2A}$ and $A_{3R}$ membrane receptors on synoviocytes, chondrocytes, and osteoblasts with a consequent increase in intracellular cAMP levels and decreased activation of NF-k$\beta$, a key regulator of the expression of metalloproteases and others pro-inflammatory factors [8]. All this translates into signals of attenuation of the inflammatory process and the promotion of both bone and cartilage regeneration.

### 1.2. Activation of Osteoinductive and Angiogenesis Pathways

Stimulation with PEMFs causes an increase in the genes of the transforming growth factor beta (TGF-$\beta$) family, in particular bone morphogenetic proteins 2 and 4 (BMPs 2–4) and the protein-mediators controlled by them [9]. In this study, it has been demonstrated that PEMFs may stimulate an early osteogenic induction in both MSCs isolated from bone marrow (BMSCs), and those derived from adipose tissue (ASCs). PEMFs act as a bioactive factor to enhance the osteogenesis of ASCs, which are an attractive cell source for clinical applications. Also, in this situation there is modulation of the Wnt/$\beta$-catenina pathway with consequent stimulation in both an osteogenic and chondrogenic sense. This is associated with an increase in the gene expression of fibroblast growth factor-2 and angipoietin-2 with improvement of the angiogenesis process [10]. PEMFs represent a noninvasive and safe strategy to modulate miRNAs with relevant roles in bone repair and with the potential to regulate the osteogenic–angiogenic coupling [11].

### 1.3. Alteration of the Extracellular Matrix

The signals taken into consideration previously play a significant role in managing the structure of the extracellular matrix (ECM), improving skeletal tissue capabilities, and facilitating the repair process [12].

The clinical implications of such interactions between biophysical stimuli and cellular responses are exploited in clinical practice to obtain tissue repair and regeneration processes in less time and with greater efficiency.

### 1.4. Ion Channels

Several studies have analyzed the action of biophysical stimulation on the cell membrane. In particular, their action at the level of signal transduction has been demonstrated. PEMFs and CCEFs control the release of $Ca^{2+}$ at the intracellular level, which, in turn, promotes the synthesis of growth factors (such as BMPs and TGF-β1) and various cell matrix proteins. In this way, biophysical stimulation promotes cell repair and mineralization [12,13].

The main fields of use currently are as follows [2]:

- Delayed union and non-union.
- Fractures with associated risk factors (open fractures, severe soft tissue damage, patient-specific factors).
- Joint-replacement surgery.
- Stress fractures.
- Bone marrow edema.
- CRPS-I.
- Cartilage repair surgery.
- Inflammatory and catabolic processes at the joint.
- Increasing bone mineral density.

In the bone-repair process an improvement in the timing of bone consolidation and fracture healing was observed [2,7,9]. For this reason, its use in recent fractures or surgical osteotomies has progressively increased in recent years.

Regarding the problem of fracture non-union, an increase in bone healing of between 73% and 85% is reported in the literature [14,15]. Even the presence of infectious processes does not compromise the validity of these treatments. Furthermore, the cost–benefit ratio supports the choice of therapy with PEMFs.

Several reviews in the literature report a valid reaction of cartilage tissue to electromagnetic biophysical stimuli [2]. Moreover, within in vivo studies it emerged that real benefits in cartilage structural improvement occur in cases of early osteoarthritis (OA) and less so in forms in which greater damage to the structural matrix is already present.

The modulation with PEMFs or CCEFs on the pathways mediated with TGF-β produces positive effects on the joint inflammatory process through an improvement in insulin growth factor-1 production in favor of anabolic processes and decreases the release of catabolic ones. In vivo, it translates into a lower production of matrix-degrading enzymes and consequent preservation of tissue components. These benefits appear particularly valid in recent and early forms of arthritis [16]. For the reasons mentioned above, the application of PEMFs after joint surgery is becoming widespread, obtaining positive results both on the tissue-healing process and on the clinical aspect such as pain relief and functional recovery [17].

The rise in the number of applications and the greater propensity to use biophysical therapy has led to a significant increase in the amount of literature on the subject.

The objective of this review is to take into consideration several recent studies regarding the application of PEMFs and Capacitively Coupled Electric Fields (CCEFs) for the improvement of bone and cartilage tissue conditions. For this reason, heterogeneous studies present in the literature were included with the aim of obtaining detailed information that can be used in a clinical context.

## 2. Application Fields

The current literature presents a valid number of publications regarding the use of magnetic fields with standardized protocols. In the first years of the application of these therapies, unfortunately, it was difficult to distinguish devices supported by real scientific evidence from more generic and unvalidated "magnetotherapies". Unfortunately, these devices are still currently available on the market with proposed indications that are often innumerable and, in truth, some lack credibility: they can range from cellulite, itching, and depression to pseudoarthritis. These are what are commonly called magnetotherapies, devices without scientific documentation, whose credibility is based simply on anecdotal evidence. We will focus on biophysical stimulation devices that are completely different from these devices based only on anecdotal evidence.

The improvement of the application protocols of these therapies, in addition to making them more effective, makes it possible to carry out an objective comparison between the different publications.

Initially we focused on the problem of bone healing, which is the first therapeutic rationale for which these therapies were applied. By broadening the focus, we were able to evaluate the results in different orthopedic and traumatology disciplines.

### 2.1. Bone Healing

Several studies in the literature support the effectiveness of biophysical stimulation with electromagnetic fields to obtain bone consolidation following a traumatic event. PEMFs' stimulation has two main areas of application. First of all, they can be use in acute fractures and, secondly, when a fracture may be at risk of non-healing due to intrinsic and/or extrinsic factors [18].

The double-blind prospective randomized study conducted by Faldini C. et al. [19] in 2010 presented valid results in patients undergoing osteosynthesis surgery with canulated screws for fractures of the femoral neck Garden 1, 2, or 3 type. The application of PEMFs is started 7 days after surgery with a treatment protocol for 8 h for 90 days. Radiographic healing was observed in 94% of treated cases vs. 69% of patients in the placebo group. Furthermore, in the placebo group, a higher percentage of osteonecrosis and higher scores in the VAS pain assessment were observed.

Similarly, a study was conducted by Assiotis A. et al. [20] in 2012 on 44 fractures involving the diaphyseal portion of the tibia with delay or non-union without ongoing infectious phenomena. Patients were treated with PEMFs for 3 h/day for a maximum of 36 weeks (average of 29.5 weeks). Fracture union was confirmed in 34 cases (77.3%). Its success is not associated with a specific fracture or patient-related variables; although, statistical significance was not demonstrated, longer treatment duration showed a trend of increased probability of union ($p = 0.081$). The author concludes that PEMFs stimulation is an effective non-invasive method for addressing non-infected tibial union abnormalities.

In 2010, Cebrián JL. et al. [21] conducted a study of tibial diaphysis fractures in non-union. In these patients, 6 months after surgery, an X-ray of absence of bone callus was observed. Clinically, they had palpation pain and preternatural mobility at the site involved. Fifty-seven patients were included in two groups. All patients had undergone endomedullary nail osteosynthesis, but only a group of 22 patients were treated with PEMFs. The remaining therapeutic protocol overlapped in the two groups. The biophysical stimulation involved the use of electromagnetic fields for 8 h/day for an average of 5.6 months. Every month, a clinical and radiographic check was performed for a minimum follow-up which lasted for 6 months. In the PEMFs group, there was recovery of 20 patients (91%) vs. 29 patients that were healed in the control group (83%). The mean time for radiographic evidence of a fracture union was 3.3 months in the PEMFs group, while it was 4.9 months in the control group ($p < 0.05$). Correlations with possible bone exposures are also considered in the results, and they are evaluated by Gustillo's classification as the age, sex, and etiology of the individual with the fracture. The authors, therefore, declare that there is no statistically significant correlation between the use of PEMFs and union of the

fracture, but it was clinically significant (relative risk 0.53, which supposes a reduction of 47% in the appearance of events). Moreover, there is a significant correlation between time to union of the fracture and the use of the stimulator and between the union of the fracture and Gustilo's classification, i.e., the greater the degree of the soft tissue injury, the lower the percentage of the union.

To facilitate the choice regarding the application of biophysical stimulation, an algorithm called FRACTING SCORE to predict bone healing in a specific body area, i.e., the tibia, was published by Massari L. et al. [22]. This study involved the recruitment of patients from 41 Italian orthopedic traumatology centers. Final targets were identified to make the evaluation homogeneous, and a 12-month follow-up was identified. Within 12 months from trauma, the date at which the fracture healed was used to calculate days and months elapsed since treatment called "healing time". At the end of the selection process, 363 patients were selected and treated with different means of synthesis (external fixation, plate and screw and nail). The purpose of the study was to assess, immediately after the treatment of the fracture, the time needed to heal.

In this way, the clinician can have objective information about the usefulness of applying the device, identifying and standardizing patients with higher risk factors and therefore, who potentially may be more suitable for these therapies.

Further confirmation of the effectiveness of stimulation with PEMFs is reported in the study of Shi HF. et al. [23] conducted on 64 patients with delayed unions of fractures involving long bones. The treatment protocol involved the application of PEMFs for 8 h a day for an average of approximately 4.5 months. The authors concluded that the early application of PEMFs for a period of 4.4–4.8 months promotes fracture healing and union rate with statistically significant results compared to the placebo group (77.4% vs. 48.1%).

To obtain a more complete evaluation of effectiveness, studies involving the application of the devices on very limited bone portions were also taken into consideration.

The study conducted by Streit A. et al. [24] on fractures of the 5th metatarsal bone took into consideration a small population of patients with delayed or failed bone union. On these patients, a bone biopsy was performed at time 0, after having performed the fixation with a cannulated screw, and a new biopsy after 3 months of stimulation with PEMFs. The authors conclude that in the group treated with biophysical stimulation, compared to the placebo, there is a significant increase in placental growth factor (PIGF), brain-derived neurotrophic factor (BDNF), and bone morphogenetic protein (BMP) 7 and 5, all of which are factors of angiogenic and osteogenic growth factors important for the formation of new bone tissue. This work presents a peculiar aspect because, unlike most studies, it does not focus exclusively on radiographic and clinical parameters but takes into consideration a more objective biomolecular variable.

In the literature, biophysical stimulation has yielded favorable outcomes in the treatment of hand fractures. We report the results of the studies conducted in 2020 by Krzyżańska L. et al. [25] and in 2023 by Factor S. et al. [26] regarding fractures of the distal radius treated with plaster casts. Both studies focus mainly on the clinical symptoms reported by patients and their adherence to physiotherapy programs. In the study by Krzyżańska L. et al. [25], PEMFs are administered from the first day for 30 min per day for a duration of 10 days, subsequently with 3 applications per week of 30 min for 6 weeks. The authors conclude that, compared to the control groups, better scores were observed on the DASH (disability of arm, shoulder, hand), SF12 (12-Item Short Form Survey), and PRWE (Patient-Rated Wrist Evaluation) questionnaires. The study by Factor S. et al. [26] expresses positive conclusions regarding the radiographic evaluation of the healing process. The group treated with PEMFs for 24 h a day had the possibility of removing the cast earlier (33 ± 5.9 days vs. 39.8 ± 7.4 days), showed a better joint ROM at both 12 and 24 weeks after the trauma, and had better scores on the clinical tests reported above.

Cheing GL. et al. demonstrated that PEMFs produce a better overall outcome in distal radius fractures [27]. In accordance with this, Lazović M. et al. used PEMP therapy during cast immobilization for Colles' fracture and observed in these patients' better results

immediately after cast removal in terms of edema and wrist range of motion (ROM) with respect to cast immobilization alone [28].

PEMFs represent a valid option even in delayed fracture healing of small bone. In fact, De Francesco F. et al. used biophysical stimulation at 8 h intervals per day for 60 days in 43 patients with delayed unions of phalanx fractures. Compared to untreated patients with PEMFs, patients undergoing treatment demonstrated higher degrees of bone growth at follow-up. Moreover, an early application of biophysical stimulation lead to a better range of motion [29].

### 2.2. Spinal Surgery

The usefulness of biophysical stimulation has also been investigated in the field of spinal surgery to improve the outcome of patients after lumbar spinal fusion surgeries. The double-blind prospective randomized multicenter study conducted by Massari L. et al. [30] in 2020 reports results relating to clinical symptoms in this area following the use of CCEFs for 90 days. In this study, forty-two patients undergoing LSF were assessed and randomly allocated to either the active or to the placebo group. Follow-up visits were performed at 1, 3, 6, and 12 months after surgery. The treatment was started 7 days after the surgery and included the application of CCEFs for 9 h a day. The device was able to stimulate the area between two intervertebral disk spaces. The Visual Analogue Scale (VAS) for pain, the 36-item Short Form Health Survey (SF-36), and the Oswestry Disability Index (ODI) for functional results were used to assess efficacy. In the active group, positive results both in the short and long term with regard to symptoms were reported by patients. For this reason, the authors conclude by advocating the usefulness of these therapies to improve quality of life.

In 2009, Massari L. et al. [31] described the main fields of use of PEMFs and their rational science considering several published studies. The study also produces an algorithm to guide the orthopedic surgeon in identifying whether the patient is suitable for biophysical stimulation and when and how to evaluate the results of stimulation. The authors have reported the usefulness of biophysical therapies in bone healing, emphasizing the importance of both the correct therapeutic indication by the clinician and the adherence by the patient. This last statement seems essential to obtain satisfactory results, so the achievement of the final goal requires correct information for the patient.

More recently, a study was conducted by Liu W. et al. [32] in 2021 on a population of 82 female patients surgically treated with percutaneous vertebroplasty for vertebral fractures due to osteoporosis. The patients selected were aged 60–75. Statistically significant differences were observed already at 1 month of follow-up regarding the Six Minutes Walking Test, while significant differences were only evident at 3 months for the VAS assessment of pain and for the radiographic appearance. The aim of the study was also to look for a correlation between clinical efficacy and post-stimulation change of bone structure; for this reason, in addition to analyzing the clinical effect, an assessment was carried out on the hip-bone level, radius, and tibia effects on bone mass and microstructure. After 1 month and 3 months, there was a marked improvement in the quality of life of the patients, but there was no increase in bone density at the level of the examined portions. It was used differently after 3 months, and there were changes to the bone microstructure.

Finally, Di Martino A. et al. concluded that PEMFs associated with multimodal management represent the gold standard in the treatment of vertebral compression fractures. This combined treatment improves patients' quality of life, promotes healing, and reduces the risks of undergoing surgery [33].

### 2.3. Joint Replacement

A further field of application for biophysical therapies concerns their application after joint replacement.

In 2015, Massari L. et al. [34] conducted a narrative review regarding the application of PEMFs in the period following hip or knee prosthetic replacement. The data present in the

literature at the time of the study demonstrated that the use of biophysical therapy could be useful to reduce bone edema, pain, and to reduce excessive bone resorption around the femoral stems.

The same conclusions were reported after a more recent review of the literature also by Lullini G. et al. [35], underlining the improvement in the clinical parameters of patients in the first post-operative period. The authors selected all available prospective studies or randomized controlled trials (RCTs) on the use of PEMFs in total joint replacement with the aim of investigating the effects of PEMFs in the post-operative period. Both reported studies highlighted the importance of these applications especially to make the patient more compliant with physiotherapy rehabilitation protocols; the authors, in fact, express positivity about the improvement of the symptomatology and about the edema and the inflammatory state of the interested portion.

In 2012, Moretti L. et al. [36] conducted a prospective, randomized, controlled study of 30 patients undergoing TKA. Patients were randomized into two groups: one group received PEMFs 4 h/day for 60 days while the second one was a control group. Both groups received the same post-operative rehabilitation program. The Knee-Score and SF-36 score were greater in the PEMFs group compared to the control one at each follow-up. Moreover, the VAS score and NSAID use was significantly reduce and joint swelling resolution was more rapid than in the controls. Adravanti P. et al. in 2014 reported similar result in 33 patients undergoing TKA: 1 month after TKA, pain, knee swelling, and functional score were significantly better in the PEMFs group compared with the control. Pain was still significantly lower in the treated group at the six-month follow-up. Three years after surgery, severe pain and occasional walking limitations were reported in a significantly lower number of patients in the treated group compared to control (7% vs. 33%) [37].

The encouraging results observed in lower limb prosthetic surgery are also reproducible for the upper limb. In 2019, the La Verde L. et al. [38] study considered the use of PEMFs for 4 h a day for 2 months following reverse shoulder arthroplasty surgery. The 50 patients considered in the study were divided into 2 groups (experimental group vs. control group) to perform a prospective randomized study. During the 6-month follow-up, statistically significant results were observed for 1, 2, and 3 months for the VAS rating scale and for the Constant score for shoulder functionality. The authors, therefore, concluded that the application of magnetic fields in this field is safe and beneficial.

In 2022, D'Ambrosi R. et al. [39] conducted a study regarding the application of PEMFs in patients treated with medial unicompartmental knee arthroplasty (UKA). The treatment protocol involved the application of PEMFs for 4 h a day, not necessarily consecutively, from day 3 to 7 post-surgery for 60 days. The following scales were used: VAS for pain, Oxford Knee Score (OKS), and Short Form 36 (SF-36) for clinical evaluation. Furthermore, the degree of knee edema and the consumption of painkillers during follow-up were evaluated. The first statistically significant results for painful symptoms were observed at 3 months, while significant clinical results were collected with OKS and SF-36 from the first month. Meanwhile, regarding edema, the first significant differences were observed at 2 months. All these results remained significant at least until the 36-month follow-up. The authors, therefore, concluded that the use of PEMFs after UKA surgery leads to a clinical improvement, to a lower consumption of drugs, to a better state of the edema, and, therefore, more generally, to greater patient satisfaction. The use of PEMFs therapy after UKA is also able to contrast the contralateral OA degeneration due to the presence of a joint inflammatory microenvironment.

### 2.4. Joint Preservation

The use of PEMFs also has a role in preserving the joint compartment. In fact, in vitro studies have obtained promising results on the possibility of cartilage repair through using electromagnetic fields [40–43].

A meta-analysis regarding the application of low- and high-frequency electromagnetic fields in the presence of inflammatory degenerative joint diseases was recently conducted

by Tong J. et al. [17]. This review states that there is evidence for low-frequency PEMFs for these pathologies at the knee and hand level, obtaining an improvement in clinical symptoms and range of motion.

The positive results for the improvement of joint inflammation and ligament lesion are also reported in the study conducted by Benazzo F. et al. [44] in patients undergoing arthroscopic knee surgery for ACL reconstruction. In total, 60 patients were identified, divided into 31 treated with biophysical stimulation and 29 in the placebo group. All patients underwent ACL reconstruction with the use of a quadruple hamstrings semitendinosus and gracilis technique. At baseline, there were no differences in the International Knee Documentation Committee (IKDC) scores between the two groups. Based on interesting pre-clinical efficacy studies, a treatment protocol was developed which involved the use of the stimulator for 4 h a day, not necessarily consecutively, for 60 days. Treatment started within 7 days from the surgery. The authors observed a lower consumption of painkillers and an improvement in the SF-36 Health Survey score at 2 and 6 months. However, it is more complex to analyze the results of the IKDC score where, as reported by the authors, there are some confounding factors. The conclusion of the work is expressed favorably regarding the reduction of the inflammatory process and the improvement of the joint catabolic processes, with benefits to both the articular cartilage and the subchondral bone. For this reason, a rationale for their use after arthroscopic surgery is suggested, particularly in those subjects who need a rapid return to physical activity.

In accordance with what has been stated, we also report the review conducted by Moretti L. et al. [45] considering works on athletes with high functional demands treated with PEMFs and extracorporeal shock wave therapy (ESWT). Even in this situation, biophysical stimulation reduced the inflammatory process, improving adherence to the physiotherapy program and the return to sporting activity in non-advanced forms of osteoarthritis (OA). The authors conclude that, although there are extremely interesting findings, it is essential to develop high-quality studies of athletes to draw stronger conclusions.

However, Zorzi C. et al. conducted a randomized prospective and double-blind study in patients with osteochondral degeneration and knee pain symptoms treated with microfractures in combination with PEMF. Patients with Grade I to IV cartilage lesions according to the Outerbridge classification were included in the study. Patients performed PEMFs for 6 h a day for 90 days. In the first month after surgery, the percentage of patients using nonsteroidal anti-inflammatory drugs (NSAIDs) was 26% in the active group and 75% in the control group ($p = 0.015$). The KOOS (Knee Osteoarthritis Outcome Score) clinical evaluation showed higher values (better joint function) in the active group at both 45 days ($73.6 \pm 10.3$ vs. $70.3 \pm 14.9$, ns) and 90 days ($83.6 \pm 7.3$ vs. $74.7 \pm 13.6$, $p < 0.05$) from surgery. At 45 days, patients in the active group already showed a level of functional recovery that patients in the placebo group would show at 90 days, demonstrating a halving of the recovery time. At 3 years of follow-up, the number of patients showing functional limitations of the knee joint was significantly higher in the control group compared to the active group (87.5% vs. 37.5%) [36]. In 2016, Reilingh ML. et al. [46] conducted a randomized multicenter trial on 68 patients after arthroscopic treatment for osteochondral defects (OCD) of the talus. During arthroscopic treatment, debridement and microfractures were performed to obtain a bone marrow stimulation. Following the surgery, a group of patients were also subjected to biophysical stimulation with PEMFs applied from the third post-operative day, for 60 days, with administration for 4 h a day. All patients performed physiotherapy rehabilitative activities with the same protocol. The aim of the study was to evaluate, compared to a placebo group, the timing of the resumption of sports and the number of patients returning to practice sports. To perform an objective clinical evaluation, the authors used the Ankle Activity Score (AAS) to evaluate, as a first outcome, the resumption of sport and the American Orthopaedic Foot and Ankle Society (AOFAS) ankle–hindfoot score, Foot and Ankle Score (FAOS), EuroQol (EQ-5D), and Numeric Rating Scales (NRSS) for pain (at rest and when running), and satisfaction was a secondary outcome. The follow-up of the patients has been extended to carry out

radiographic evaluations after 12 months, in order to compare the clinical results with the radiographic ones. The authors affirm that the application of PEMFs has not shown an improvement in the timing and the resumption of sport and the radiographical aspect, in this context, and that statistically significant differences have not been observed compared with the placebo group which also prolonged the evaluation to 12 months. Although the results showed no difference between the treated and control groups in the number of patients who returned to sport 1 year after surgery, in the PEMFs group 96% of patients returned to sport by week 30, while in the control group only 74% did ($p = 0.017$). In the control group, the same percentage as the PEMFs group (96%) was achieved at week 52.

In another clinical trial on patients with osteochondral lesions of the talus, treated using graft transplantation with the addition of bone marrow concentrate (BMDC) in a single operating session (ONE-STEP Method), Cadossi et al. showed there was less pain in the experimental group with PEMFs at 2, 6, and 12 months of follow-up, and there were significantly higher functional results in the group of stimulated patients compared to the controls. Early rehabilitation with less pain can certainly lead to a better clinical outcome even after a long time [47].

Similar results were also found in a group of patients undergoing autologous chondrocytes transplantation in the presence of scaffolds (MACI) and who were treated with PEMFs. At the baseline, the two groups were perfectly comparable for clinical scores and cartilage injury characteristics. The International Knee Documentation Committee (IKDC) score showed a significant improvement in the treated group compared to the control at 1- ($p = 0.01$), 2- ($p = 0.041$), and 60 ($p = 0.001$)-month follow-ups. In addition, there was a statistically significant difference between the groups at 1 month ($p = 0.023$) and 60 months for SF-36 ($p = 0.006$) and at 60 months for EuroQol ($p = 0.020$). A significant reduction in pain was observed in the treatment group compared to the control at 1 ($p = 0.018$), 2 ($p = 0.043$), and 60 ($p = 0.011$) months of follow-up [48].

Clinical results from all studies show that the cartilage regeneration and/or repair method associated with PEMFs is an effective solution for the treatment of chondral and osteochondral lesions of the joint in the field of regenerative medicine and tissue engineering.

Similar results are reported by Massari L. et al. [2] in the narrative review conducted in 2019. This publication examined the results in various specialist fields, concluding positively regarding the application of biophysical therapies. The study examines the different fields of use to draw indications from the literature. Improvements in the bone-healing process have been observed in subjects at risk of malunion, along with improvements in the symptoms reported in patients undergoing spinal surgery, better rates of osseointegration in the femoral components of cementless hip prostheses, effective results of core decompression and grafts of trabecular bone in the treatment of osteonecrosis of the femoral head, and improvements in inflammatory processes in the articular compartment. None of the authors of these studies suggest a generalized use of biophysical stimulation, but the importance of a reasoned therapeutic prescription is highlighted.

In 2013, Nelson R. et al. recruited 34 patients with early knee osteoarthritis in a randomized, placebo-controlled, double-blind pilot clinical study. Results showed that the VAS pain score decreased in the active cohort by $50 \pm 11\%$ versus the baseline, starting at day 1 and persisting to day 42 ($p < 0.001$). The authors concluded that non-invasive PEMFs therapy can have a significant and rapid impact on pain from early knee osteoarthritis [49]. Similar results were also obtained by other authors [50–52].

In 2006 Massari L et al. retrospectively analyzed 76 hips in 66 patients with osteonecrosis of the femoral head. Patients with Ficat stage I, II, or III osteonecrosis of the femoral head were treated with pulsed electromagnetic field stimulation for eight hours per day for an average of five months. They found that PEMFs preserved 94% of Ficat stage-I or -II hips, recommending this treatment in the early stages of osteonecrosis of the femoral head [53].

Similar results were obtained by Marcheggiani Muccioli GM. et al. The authors found good to excellent clinical results in 75% of patients and protected 85.7% of knees with spontaneous osteonecrosis of the Koshino stage I knee from prosthetic surgery at the 24-month follow-up [54].

Bagnato GL et al. analyzed 60 patients with knee osteoarthritis and persistent pain. After 1 month, a mean effect of treatment with PEMFs of $-0.73$ (95% CI $-1.24$ to $-0.19$) was observed in the VAS score, while the effect size was $-0.34$ (95% CI $-0.85$ to $0.17$) for the WOMAC score. In total, 26% of patients in the PEMFs group discontinued NSAID/analgesic therapy. No adverse events were observed [55].

In contrast, Ozgüçlü E. et al. found that PEMFs does not have an additional effect on the classical physical treatment for reducing symptoms of knee osteoarthritis. However, in this study, PEMFs were used for 30 min per day, thus reducing any beneficial effects [56].

Finally, a recent narrative review found that PEMFs can improve symptoms and the function of joints, such as the knee, in patients with non-advanced osteoarthritis [57]. Viganò M. et al. in a recent review confirmed that PEMFs therapies are safe and effective treatments for the control of knee OA-related pain and disability in the short term [58].

In conclusion, PEMFs have proven to be a valid option for reducing pain, controlling inflammatory process, favoring functional recovery, and improving the quality of life of patients. Encouraging results have been obtained in preclinical studies, as well as on cartilage regeneration after surgery. There is a lack of knowledge of the effectiveness on cartilage regeneration.

*2.5. Osteopenia and Osteoporosis*

Osteopenia and osteoporosis are diseases characterized by a decrease in bone mineral density, leading to a weakening of the bones and an increased risk of fractures. Recently, the scientific community has increasingly begun to highlight the effectiveness of PEMFs in the treatment of osteopenia and osteoporosis [59].

In 2019, Wang P. et al. found that PEMFs inhibit osteoclast formation by regulating the ratio of RANKL/OPG [60].

In 2013, LIU HF. et al. randomly treated 44 patients with alendronate or PEMFs. The primary endpoint was the mean percent change in lumbar spine bone mineral density (BMDL), while the secondary endpoints were mean percent change in left proximal femur bone mineral density (BMDF), serum 25OH vitamin D3 (25(OH)D) concentrations, lower limb manual muscle test (LE MMT) score, and Berg balance scale (BBS) score. Using a mixed linear model, there was no significant treatment difference between the two groups in the BMDL, BMDF, total LE MMT score, and BBS score ($p \geq 0.05$). For 25(OH)D concentrations, the effects were also comparable between the two groups ($p \geq 0.05$) with the Mann–Whitney's U-test. These results suggested that a course of PEMFs treatment with specific parameters was as effective as alendronate in treating post-menopausal osteoporosis within 24 weeks [61].

Similar results were obtained by other authors [62,63]. A recent review showed positive effects of PEMFs in the treatment of osteoporosis in 23 out of 24 studies included in the study. In particular, PEMFs increase bone mineral mass and reduce symptoms related to osteoporosis [64].

A narrative review conducted by Zhang W. et al. concluded that electrical stimulation represents a good, non-invasive, and effective treatment for osteoporosis. The use of PEMFs is the treatment method with the greatest evidence of success. However, the specific treatment parameters of frequency and treatment time are not yet conclusive [65].

**3. Conclusions**

The analysis of the recent literature determines the amount of evidence regarding the improvement of the bone-healing process in patients treated with PEMFs. The effectiveness of these therapies is now consolidated with cure rates that largely support the cost–benefit ratio. In the literature review we conducted, no validated scientific studies were identi-

fied that reported side effects of these therapies. All the works considered reiterate the fundamental importance regarding the correct diagnosis of use, methods, and times of application, and above all compliance with the treatment by the patient. The dominant opinion that emerges is in favor of the use of stimulation with electromagnetic fields in the sectors taken into consideration. The studies in which there was no evidence of statistically significant results presented, as declared by the authors, bias with potential impact on the interpretation of the result.

As observed in the studies reviewed in this discussion, there are different devices that deliver different combinations of physical parameters, but all of them give a positive therapeutic result.

Our goal is not to compare the results obtained with different biophysical stimulation devices and identify the best one, but it is to be able to recommend to our patients a biophysical stimulation device, free of side effects or contraindications, based on scientific support and documented clinical validation.

In the presence of osteoarthritis or osteonecrosis, the choice of the moment of application of the stimulation plays a fundamental role; better results have in fact been reported in recent forms of OA with a predominance for the inflammatory component, while the results are less evident in advanced and structured forms of joint damage. The effectiveness on the joint inflammatory phase makes the application interesting after joint replacement and in the field of arthroscopic surgery due to promising results regarding the inhibition of catabolic processes. We also found positive impacts on clinical symptoms and patient compliance with rehabilitation therapies. For this reason, notable applications can be expected in the fields of prosthetic surgery and sports medicine. In the coming years, we will observe a strong increase in applications in these areas. In fact, the valid results on joint inflammatory pathology demonstrate a strong indication for use both in the field of joint replacement and joint preservation. In this last sector, and even more so in sports medicine, it will be extremely interesting to evaluate the results emerging from preclinical studies on devices dedicated to the regeneration of cartilage tissue. In fact, unlike bone tissue, there is currently no dominant opinion on the regeneration of cartilage tissue; however, the preliminary results mentioned above suggest an optimistic attitude in this field of application.

We, therefore, conclude by stating that biophysical therapy can provide valuable aid in clinical practice if the correct indications and methods of administration are respected.

## 4. Future Directions

The use of biophysical stimulation in the field of bone healing is now a consolidated reality; however, the need to have well-defined recommendations emerges to optimize the indications of use. It would be advisable to increase the multicenter studies in order to have larger patient populations, selected with similar criteria, and subjected to overlapping treatment protocols and devices.

Objective assessments, such as the *FRACTING SCORE* [12], capable of predicting the risk of malunions, would be precious tools in the hands of clinicians; therefore, we hope that further objective assessments will be developed.

The collection of evidence will be the basis for the wider use of these therapies and will also serve to improve patients' adherence to the rehabilitation process both in the traumatology and elective surgery fields.

In the literature, there are no side effects reported following the therapeutic protocols applied. For this reason, it would be interesting to develop therapeutic strategies for a chronic patient. In fact, the possibility to perform the administration locally makes this stimulation particularly suitable for these situations. In this way, the full potential of PEMFs and CCEFs in positively modulating harmful processes could be explored. This approach could also further justify the initial cost to patients in some situations.

Another area of interest is where data are used for examining the application on tissues involved in infectious processes. There are both clinical studies in patients with traumatic

injuries and concomitant infectious processes and preclinical studies whose results show a potential role both in improving tissue condition and as a direct bactericidal effect.

That is why we hope that there will be further work in this area to gather data that can be accessed quickly.

Further stimuli come from the results observed in the field of cartilage tissue regeneration. We will probably see a remarkable development of the literature in this regard due to very promising in vitro results. It will certainly be necessary to carry out a careful selection of the target population to avoid incorrect interpretations of the results; however, in subjects that still have valid regenerative potential, a valid rational use could be configured.

Finally, we must not forget that the results in all the areas considered are closely linked to the quality of the signal generated and that in the literature, as well as in the market, you can find many devices with different technical characteristics, and it is this quality parameter that plays a key role in the final result. For this reason, only continuing with a critical approach, in their various fields of use, and with a scrupulous collection of clinical data will it be possible to completely understand the potential of these therapies.

**Author Contributions:** Conceptualization, G.C. and L.M.; methodology, A.S.; data curation, A.S., E.G. and S.L.; writing—original draft preparation, A.S., S.S., E.G. and S.L.; writing—review and editing A.S., S.S. and G.C.; supervision, L.M. All authors have read and agreed to the published version of the manuscript.

**Funding:** This research received no external funding.

**Institutional Review Board Statement:** Not applicable.

**Informed Consent Statement:** Not applicable.

**Data Availability Statement:** The data presented in this study are available on request from the corresponding author.

**Conflicts of Interest:** S.S. is an employee of IGEA SpA—Clinical Biophysics. The other authors declare no conflicts of interest.

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
