# Peer review of "Pulsed Electromagnetic Field Stimulation in Bone Healing and Joint Preservation: A Narrative Review of the Literature"

_applsci, doi:10.3390/app14051789_

Round 1
Reviewer 1 Report
Comments and Suggestions for Authors
The authors present an interesting narrative review of the literature concerning the use of Pulsed Electromagnetic Fields (PEMFs) in bone healing process and on joint preservation. The information given indicates that the use of biophysical therapies is now important in several areas of medical practice, such as orthopedics, traumatology, and sports medicine, among others. The work is interesting and adds knowledge in the field. I have a few comments.
1. Please in page 3, lines 118-128, please organize the text in the correct part of the manuscript, it seems to be part of the introduction and not of section 1.4.
2. I suggest including some figures to explain the effect of (PEMFs) on Bone and cartilage.
Comments on the Quality of English LanguageMinor editing of English language required
Author Response
Dear editors,
in the revision presented the introductory part has been modified, improving the technical aspect. Furthermore, greater importance was given to the discussion, presenting further relevant articles and discussing these works more clearly. For this reason we considered it appropriate to increase the works taken into consideration, in order to obtain better coverage of the topics covered. Finally we have increased the final part relating to future developments. I hope you can appreciate these changes!

Reviewer 2 Report
Comments and Suggestions for Authors
This review looks good to this reviewer. The addition of some discussion of the mechanisms by which the fields effect the processes would add to the value of the paper. In particular, the effectives of the magnetic fields on chemical reactions rates is not often included. Other work shows that static low frequency magnetic and radio frequency fields can change the growth rates other cells, the metabolic rates and the concentrations of radicals. Electric field move ions around and to change chemical reactions rates by changing collision times. Differences between the CCEF and the pulsed fields in effectiveness would worth discussing.
Author Response

(The authors gave the same response as above.)

Reviewer 3 Report
Comments and Suggestions for Authors
This review explores the growing use of biophysical stimulation therapies, specifically Pulsed Electromagnetic Fields (PEMFs) and capacitively coupled electric fields (CCEFs), in orthopedics and traumatology. Analyzing 17 studies from the last 15 years, emphasizes the effectiveness of these therapies in bone healing, inflammation, and patient compliance. While positive outcomes are noted in joint compartments, evidence for cartilage regeneration is lacking. The paper concludes that PEMFs and CCEFs are beneficial in orthopedics, prosthetic surgery, and sports medicine, highlighting the importance of accurate diagnosis, proper application, and patient adherence for successful outcomes, with no reported side effects.
The review is important but lacks some key information and needs significant improvements.
1. What is the novelty of this review? And why it is important? This needs to be highlighted in the abstract.
2. The introduction section needs to be improved. It is advised to add some literature regarding how electromagnetic fields interact with biological systems for new readers. The addition of the following recent article (https://doi.org/10.3390/ijms23169288) might be a useful source for authors to improve the background information.
3. For a full review article, the reference number should be more than 100. The authors should improve the literature review.
4. What specific clinical conditions within orthopedics and traumatology demonstrate the highest efficacy when treated with Pulsed Electromagnetic Fields (PEMFs) and capacitively coupled electric fields (CCEFs)?
5. Most of the references cited in this review are outdated, authors are encouraged to cite recent advancements in their study topic.
6. Include some graphical representation or some figure to make your review attractive and more informative. Relying on text only might not be so attractive for readers.
7. The author should prepare a table to summarize the key literature review in this manuscript.
8. The conclusion is too long and does not seem so informative. Most of the information is introduction. Revise it.
9. In the future direction, I recommend highlighting only key messages, not background information. Also, the statement in line 611, “you can find many devices with different technical characteristics” is not appropriate.
10. The manuscript contains several typos and grammatical errors. I recommend thorough proofreading to identify and correct these issues.
Comments on the Quality of English LanguageThe manuscript contains several typos and grammatical errors. I recommend thorough proofreading to identify and correct these issues.
Author Response

(The authors gave the same response as above.)

Round 2
Reviewer 3 Report
Comments and Suggestions for Authors
I was disappointed to find that the authors ignored the reviewers' comments and the manuscript was not modified accordingly. I do not see a detailed response to the reviewers's comments. This manuscript cannot be accepted unless the questions posed in round 1 of the reviews are explicitly addressed.
Also, the response letter attached by the authors is not the correct file.
English improvement required.
Author Response

(The authors gave the same response as above.)

Round 3
Reviewer 3 Report
Comments and Suggestions for Authors
I cannot locate the revisions in the manuscript corresponding to my comments, and there is no text that has been highlighted. It is recommended that the author clearly highlight the changes in the revised manuscript and provide a detailed point-by-point response to the comments in the response letter.
Comments on the Quality of English LanguageEditors are requested to take note that the changes in the revised manuscript are not highlighted, making it difficult for me to identify changes based on my comments from the first round. Additionally, the file attached to the response letter appears to be incorrect. There is not point-by-point response to my comments.
Round 4
Reviewer 3 Report
Comments and Suggestions for Authors
The paper seems improved in this version. I recommend to accept the paper in its present form.